# Improved Variational Bayesian Phylogenetic Inference using Mixtures

**Ricky Molén**[*]                                                    *rickym@kth.se*
*KTH Royal School of Technology*
*Science for Life Laboratory*

**Oskar Kviman**[*]                                                  *okviman@kth.se*
*KTH Royal School of Technology*
*Science for Life Laboratory*

**Jens Lagergren**                                                   *jensl@kth.se*
*KTH Royal School of Technology*
*Science for Life Laboratory*

**Reviewed on OpenReview:** *https://openreview.net/forum?id=TBLMrHaFFH*

## Abstract

We introduce VBPI-Mixtures, an algorithm aimed at improving the precision of phylogenetic posterior distributions, with a focus on accurately approximating tree-topologies and branch lengths. Although Variational Bayesian Phylogenetic Inference (VBPI)—a state-of-the-art black-box variational inference (BBVI) framework—has achieved significant success in approximating these distributions, it faces challenges in dealing with the multimodal nature of tree-topology posteriors. While advanced deep learning techniques like normalizing flows and graph neural networks have enhanced VBPI's approximations of branch-length posteriors, there has been a gap in improving its tree-topology posterior approximations. Our novel VBPI-Mixtures algorithm addresses this gap by leveraging recent advancements in mixture learning within the BBVI domain. Consequently, VBPI-Mixtures can capture distributions over tree-topologies that other VBPI algorithms cannot model. Across eight real phylogenetic datasets and compared to the considered benchmarks, we show that VBPI-Mixtures result in lower-variance estimators of the marginal log-likelihood and smaller KL divergences to an MCMC-based approximation of the true tree-topology posterior.

## 1 Introduction

Phylogenetic inference has a wide range of applications in various fields, such as molecular evolution, epidemiology, ecology, and tumor progression, making it an essential tool for modern evolutionary research. Bayesian phylogenetics allows researchers to reason about uncertainty in their findings about the evolutionary relationship between species.

The posterior distribution over phylogenetic trees given the species data is, however, challenging to infer, since the latent space is a Cartesian product of the discrete tree-topology space and the continuous branch-length space. Furthermore, the cardinality of the tree-topology space grows as a double factorial of the number of species (taxa), making the marginal likelihood computationally intractable in most interesting problem settings.

For over two decades, Markov Chain Monte Carlo (MCMC) approaches have been the go-to approaches for Bayesian phylogenetic analysis, where the MrBayes software (Huelsenbeck & Ronquist, 2001) has been

---
[*]Equal contribution.

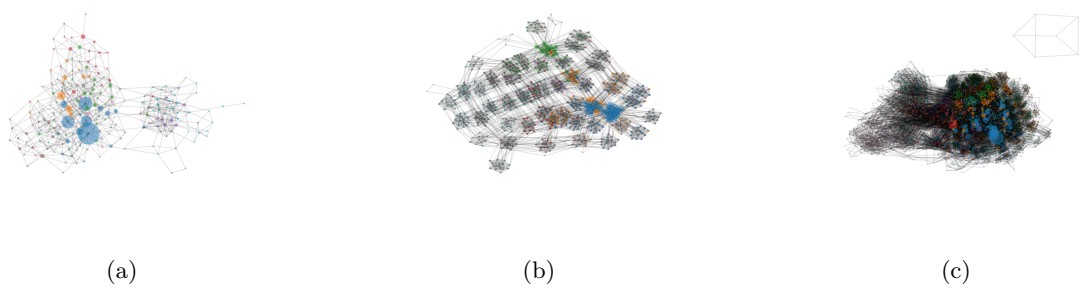

|        |        |        |
|:------:|:------:|:------:|
| (a)    | (b)    | (c)    |

Figure 1: Visualization of samples from the tree-topology posterior using a 1,000,000,000 iterations long MCMC run on (a) DS4, (b) DS7 and (c) DS8. Nodes represent unique tree-topologies and are colored based on cluster assignments, illustrating the multimodality of the tree-topology posterior. The size of a node is determined by its sampling frequency, which is why nodes with low frequency appear black. More details in Sec. 2.

particularly popular. However, random walk Metropolis-Hastings MCMC methods (Huelsenbeck & Ronquist, 2001; Höhna et al., 2016) rely on local operations to explore the tree-topology posterior, a limitation which is known to require long MCMC runs in order to visit posterior regions which are separated by low-probability tree topologies (Whidden & Matsen IV, 2015). Sequential Monte Carlo methods for Bayesian phylogeny (Bouchard-Côté et al., 2012; Wang et al., 2015; 2020) have been proposed to avoid these local operations, but the resampling mechanism can filter out important trees in early steps of the algorithm as well as cause degeneracy, necessitating many particles.

More recently, variational inference (VI) has been applied to Bayesian phylogenetics. In general, VI is often promoted over sampling-based approaches in high-dimensional problems as a variational approximation of the posterior is obtained from optimization, making VI less vulnerable to the curse of dimensionality. However, in practice, it can be challenging to do VI without utilizing sampling. For example, in Koptagel et al. (2022) coordinate-ascent update equations are derived in the phylogenetic setting, but these are evaluated using importance sampling.

In black-box VI (Ranganath et al., 2014), the gradients are instead taken of Monte-Carlo integrated estimates of the objective, typically the evidence lower bound (ELBO), using samples taken from the variational approximation. This approach has been successfully applied in the Variational Phylogenetic Bayesian Inference (VBPI; Zhang & Matsen IV (2019)) framework, along with its extensions (Zhang, 2020; 2023). In these extensions, more complicated branch-length approximations have been proposed using normalizing flows (NFs; Rezende & Mohamed (2015)) and graph-neural nets (GNNs; Kipf & Welling (2017)).

BBVI allows the practitioner to learn posterior approximations without deriving update equations or closed-form gradient formulations, the samples are taken from a distribution that is commonly known to concentrate on high-probability regions of the posterior, resulting in a learning procedure that does not sufficiently explore the posterior distribution. This was addressed for continuous distributions in Ruiz et al. (2016), where samples were instead drawn from an overdispersed proposal distribution. In discrete hierarchical models, similar to the targets approximated by VBPI, however, insufficient exploration may result in low-level states not being properly modeled. As the tree-topology posterior is known to typically be multimodal with many "subpeaks" (Whidden & Matsen IV, 2015) (visualized in Fig. 1), it is thus of significant importance to encourage the tree-topology approximation to explore the posterior.

We propose VBPI-Mixtures, a novel combination of two recent advances, VBPI and mixture learning in BBVI (Kviman et al., 2023). The mixture components cooperatively explore the tree-topology posterior during learning, and the increased flexibility of the VBPI-Mixtures results in approximations that can model posteriors intractable for the vanilla VBPI (see Sec. 3). Using a toy experiment where we design a complicated hierarchical categorical target distribution and pick a variational family such that it does not contain the target distribution, we first show that the mixtures of these sub-optimal variational distributions achieve

smaller Kullback-Leibler divergences to the targets and smaller importance-weight variances than a single-component approximation which uses more Monte Carlo samples to estimate the objective. We then apply VBPI-Mixtures on eight popular real datasets, visualize the learned tree-topology approximations and compare numerically with MCMC "golden runs", illustrating the joint exploration of the tree-topology space by the mixture components. Additionally, we derive a mixture-specific formulation of the VIMCO estimator (Mnih & Rezende, 2016)—a variance reduction technique using control variates to estimate the gradient of the importance weighted ELBO (Burda et al., 2015). Our contributions can thus be summarized as follows:

- We propose VBPI-Mixtures, a novel algorithm for Bayesian phylogenetics.

- We show that mixtures of subsplit Bayesian nets (SBNs) can approximate distributions that a single SBN cannot, making a persuasive case for VBPI-Mixtures.

- We derive the VIMCO estimator for mixtures.

- We visualize a two-component mixture of SBNs on real data, verifying that the components jointly explore the tree-topology space.

- Experimentally, we achieve state-of-the-art results on eight popular real phylogenetics datasets in terms of producing lower-variance estimators of the marginal log-likelihood, showing that mixtures of SBNs can accurately capture tree-topology posteriors that non-mixture SBNs cannot, and produce KL divergences to an MCMC-based approximation of the true tree-topology posterior that decrease with $S$.

## 2 Background

Let $\mathcal{B} = \{b(e) : e \in E(\tau)\}$ denote the set of branch lengths for a topology, $\tau$ and $E(\tau)$ is the set of edges in $\tau$. Furthermore, the data, $X = \{X_1, ..., X_N\} \in \Omega^{M \times N}$, are an observed sequences of length $M$ on the $N$ leaves of the phylogenetic tree. Each entry in $X_{m,n}$ is a character in the alphabet $\Omega$, e.g., $X_{m,n} \in \{A, C, G, T\}$ if DNA sequences are considered. The posterior distribution over leaf-labeled phylogenetic trees,

$$p(\mathcal{B}, \tau | X) = \frac{p(X | \tau, \mathcal{B}) p(\mathcal{B} | \tau) p(\tau)}{p(X)}, \tag{1}$$

is intractable due to its normalizing constant, $p(X)$, i.e., the marginal likelihood.

Although $p(X)$ is intractable, the three terms in the generative model in Eq. 1 can be computed efficiently. Typically, $p(\tau)$ is a uniform distribution over the (rooted or unrooted) tree-topology space, and the branch-length prior is an exponential distribution with rate $\lambda$, such that $p(\mathcal{B} | \tau) = \prod_{e \in E(\tau)} \lambda e^{-\lambda b(e)}$. The likelihood, $p(X | \tau, \mathcal{B})$, can be evaluated in linear (in $N$) time using the standard dynamic programming algorithm proposed by Felsenstein (2003).

Let a *clade* be a non-empty subset of the set of the $N$ leaf labels, $\mathcal{X}$. A *subsplit* is a partition of this clade into two lexicographically ordered, disjoint clades, while a *split* is simply a root subsplit—a bipartition of $\mathcal{X}$. Furthermore, a primary subsplit pair (PSP) is a subsplit conditioned on a split—a tripartition of $\mathcal{X}$. In Appendix A we additionally give a brief introduction to Bayesian phylogenetic inference for machine learning researchers.

**Variational Inference in Bayesian Phylogenetics** The VI-based approach to Bayesian phylogenetics is to approximate $p(\mathcal{B}, \tau | X)$ using a simpler distribution, $q_{\psi,\phi}(\mathcal{B}, \tau) = q_\psi(\mathcal{B} | \tau) q_\phi(\tau)$. Generally, in VI, the approximations are learned by maximizing the evidence lower bound (ELBO),

$$\mathcal{L}(X) = \mathbb{E}_{q_{\psi,\phi}(\mathcal{B}, \tau)} \left[ \log \frac{p(X | \tau, \mathcal{B}) p(\mathcal{B} | \tau) p(\tau)}{q_\psi(\mathcal{B} | \tau) q_\phi(\tau)} \right], \tag{2}$$

implicitly minimizing the reverse Kullback-Leibler (KL) divergence from $p(\mathcal{B}, \tau | X)$ to $q_{\psi,\phi}(\mathcal{B}, \tau)$.

**Subsplit Bayesian Networks**   Given a set of tree topologies, $\mathcal{T}$,[1] it is straightforward to form a look-up table of all subsplits in $\mathcal{T}$. The SBN uses the look-up table to define support over possible tree topologies and learns the probabilities of the subsplits in the table via, for example, stochastic optimization. As the look-up table contains probabilities of subsplits, it is referred to as a *conditional probability table* (CPT). When the CPT has been learned, the SBN provides a tractable probability distribution over tree topologies from which it is possible to sample. See Zhang & Matsen IV (2018) or Zhang & Matsen IV (2024) for the original, more in-depth accounts of SBNs.

Regarding the pre-computed set of tree topologies, $\mathcal{T}$, that are given as input to the SBNs, it is worth noting that, although $\mathcal{T}$ is a subset of the set of all possible tree topologies, this should not be considered as a major constraint on the support of the SBNs. Support for this argument has been provided in Zhang & Matsen IV (2024) (see Fig. 5 in the reference), where they showed that the number of pre-computed trees is close to the total number of topologies in the support of a 10 billion MCMC (MrBayes (Huelsenbeck & Ronquist, 2001)) run. Also, the topology-posterior coverage of UFBoot is close to 100% of the MrBayes topology posterior.

**VBPI**   In VBPI (Zhang & Matsen IV, 2019), the posterior approximations are learned by maximizing a multi-sample (Burda et al., 2015) version of $\mathcal{L}(X)$,

$$\mathcal{L}(X; K) = \mathbb{E}_{\mathcal{B}^k, \tau^k \sim q_{\psi, \phi}(\mathcal{B}, \tau)} \left[ \log \frac{1}{K} \sum_{k=1}^{K} \frac{p(X|\tau^k, \mathcal{B}^k) p(\mathcal{B}^k|\tau^k) p(\tau^k)}{q_\psi(\mathcal{B}^k|\tau^k) q_\phi(\tau^k)} \right], \tag{3}$$

where $q_\phi(\tau)$ is an SBN with a learnable CPT, $\phi$, and $q_\psi(\mathcal{B}|\tau)$ is a multivariate LogNormal distribution with a diagonal covariance matrix, such that

$$q_\psi(\mathcal{B}|\tau) = \prod_{e \in E(\tau)} q(b(e)|\mu(e, \tau), \sigma^2(e, \tau)). \tag{4}$$

Two different parameterizations of $\mu(e, \tau)$ and $\sigma(e, \tau)$ have previously been proposed. The simpler approach is to let $\mu(e, \tau) = \psi_{e/\tau}^\mu$ and $\sigma(e, \tau) = \psi_{e/\tau}^\sigma$, where $e/\tau$ denotes a split of $\tau$ in edge $e$. The parameters $\psi_{e/\tau}^\mu$ and $\psi_{e/\tau}^\sigma$ are shared among all tree topologies where $e/\tau$ exists, resulting in an amortized mapping from a tree topology to the parameters of $q_\psi(\mathcal{B}|\tau)$. Additional local information about the given $\tau$ can be added into the parameterization of the approximation by using PSPs,

$$\mu(e, \tau) = \psi_{e/\tau}^\mu + \sum_{i \in e /\!\!/ \tau} \gamma_i^\mu, \quad \sigma(e, \tau) = \psi_{e/\tau}^\sigma + \sum_{i \in e /\!\!/ \tau} \gamma_i^\sigma, \tag{5}$$

where $e /\!\!/ \tau$ denotes the set of PSPs neighboring to the split $e/\tau$, and $\gamma_i^\mu$ is a learnable parameter associated with the $i$-th pair.

**Multimodality of the tree-topology posterior**   In Whidden & Matsen IV (2015), modes are referred to as clusters of MCMC samples that are densely grouped in the tree-topology space and have high posterior density compared to their neighbors. They proposed a method for detecting and quantifying peaks by calculating the reversible subtree pruning and regrafting distance between topologies, combined with the MrBayes MCMC posterior probability. By applying this method, it was identified that certain datasets had a high number of modes (e.g., DS1, DS4, DS5, DS6, DS7), which shows the complexity of the tree-topology space. In Fig. 1 we visualize the multimodality of the posterior on three datasets.

**Mixtures in Black-Box VI**   Learning mixtures of approximations in (black-box; Ranganath et al. (2014)) VI (Nalisnick et al., 2016; Morningstar et al., 2021; Kviman et al., 2023) is a compelling off-the-shelf technique to increase the flexibility of a variational approximation. Mixtures can be applied to any variational approximation, including an NF-based approximation or one for discrete latent variables, with little overhead.

The objective function, the ELBO for mixtures, is estimated by sampling from each mixture component in a stratified manner (Morningstar et al., 2021), or via multiple importance sampling techniques (Kviman et al.,

---

[1]$\mathcal{T}$ is in practice obtained from some efficient tree-topology sampling algorithm, typically UFBoot (Minh et al., 2013).

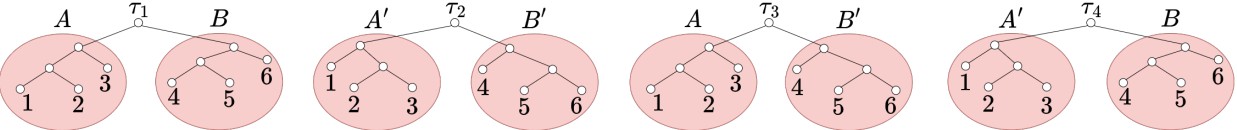

Figure 2: Mixtures of SBNs increase the flexibility of the tree-topology approximation. For instance, they can exactly capture a target distribution that assigns its probability uniformly across $\tau_1$ and $\tau_2$, leaving zero probability to $\tau_3$ and $\tau_4$. Meanwhile, this is not possible for a single SBN. See Example 3.1 for details.

2022), why the objective is often referred to as *MISELBO*. Maximizing MISELBO encourages the mixture components to cooperatively cover the target distribution, which is thought to be the key ingredient for their success in density estimation tasks (Kviman et al., 2023). In the next section, we formulate MISELBO for VBPI and explain why mixtures of SBNs are beneficial for exploring the complex tree-topology space.

Importantly, the MISELBO objective shares similarities with the objective used in the mixture of experts (MoE) literature (Shi et al., 2019), however, note that in MoEs the mixture components are conditioned on different data types (e.g., images or texts), whereas here the data used for conditioning is identical for all components. As such, there is a crucial distinction between the MISELBO objective and the one from the MoE literature.

**ARTree**  In Xie & Zhang (2023), the authors employ an autoregressive sampling where actions are sampled of where a leaf should be inserted into a graph starting with three leaves. When all leaves in the taxa are inserted they have sampled one topology. The model utilizes a fixed order of insertions such that the topology can only be achieved with a single unique order of decisions of where the insertions of leaves happen. Additionally, ARTree is an alternative to SBNs by leveraging GNNs to encode topological features, thereby eliminating the constraint of pre-sampled topologies inherent to SBNs.

**PhyloGFN**  PhyloGFN Zhou et al. (2024) employs deep generative flow networks Bengio et al. (2023) to sample topologies based on a set of actions of joining leaves and subtrees. The generative flow network setup makes it possible to handle the non-unique set of actions to sample the topology as opposed to the constraint used in Xie & Zhang (2023). This model constructs phylogenetic trees using a bottom-up approach with transformer networks and is not constrained by pre-sampled topologies.

## 3  Variational Bayesian Phylogenetic Inference using Mixtures

Here we present our proposed method, VBPI-Mixtures. We derive the VIMCO estimator for learning mixtures of SBNs, and show how to combine mixtures of branch length models with an expressive NF model. We start by providing the MISELBO formulation for VBPI with $K$ importance samples,

$$\mathcal{L}(X; K, S) = \frac{1}{S} \sum_{s=1}^{S} \mathbb{E}_{\mathcal{B}_s^{1:K}, \tau_s^{1:K} \sim q_{\psi_s, \phi_s}(\mathcal{B}, \tau)} \left[ \log \frac{1}{K} \sum_{k=1}^{K} \frac{p(X|\tau_s^k, \mathcal{B}_s^k) p(\mathcal{B}_s^k|\tau_s^k) p(\tau_s^k)}{\frac{1}{S} \sum_{j=1}^{S} q_{\psi_j}(\mathcal{B}_s^k|\tau_s^k) q_{\phi_j}(\tau_s^k)} \right]. \quad (6)$$

To evaluate $\mathcal{L}(X; K, S)$, we approximate the $s$-th expectation by Monte-Carlo integration using simulations from $q_{\psi_s, \phi_s}(\mathcal{B}, \tau)$. Although there exist works where the mixture component weights are learned (Roeder et al., 2017; Morningstar et al., 2021), we constrain ourselves to uniform mixture weights as in Kviman et al. (2023).

**Mixtures promote exploration**  Note that maximizing Eq. (6) w.r.t. the variational parameters will promote the components to diversify and explore regions of the latent space that correspond to high-probability regions measured by $p$. That is, samples drawn from component $s$ that result in high values in the numerator will indicate that the $s$-th component covers a part of the latent space corresponding to a high-probability region, attracting the other components to this region, while the denominator penalizes similarities among

the components, promoting the other components to spread out and find other important parts of the latent space. This exploratory behavior is crucial in black-box VI as the ELBO is only evaluated in sampled (visited) latent variables (states).

More specifically, the samples are proposed from the same distribution that attempts to maximize the ELBO. Consequently, for $S = 1$, there is a risk that, for a fixed $K$, less probable regions will not be sufficiently explored during learning and thus will be poorly modeled. In fact, Zhang & Matsen IV (2019) showed that the vanilla VBPI does not necessarily benefit from more importance samples during training in terms of resulting marginal log-likelihood scores. Fortunately, the MISELBO objective offers a promising solution, as the mixture components are promoted to spread out and efficiently explore the multimodal phylogenetic posterior.

**Mixtures increase the flexibility of the approximations**  A mixture of LogNormal pdfs is clearly more flexible than a single LogNormal pdf. For mixtures of SBNs, this is also true but may be less clear. An SBN constructs a tree by stochastically partitioning the available clades. The partition of a clade is sampled *independently* of the partitions sampled in the other clades. Mixtures of SBNs allow for modeling correlations in the sampling of the partitions, and thus increase the flexibility of the approximation. We explain this feature with a simple example, which trivially generalizes to larger trees and also applies to unrooted trees.

**Example 3.1** *Consider the four rooted tree topologies in Fig. 2 where $A, B, A'$ and $B'$ are four different subtrees for the two clades (in red), and a target density, such that $p(\tau_1) = p(\tau_2) = 0.5$. A uniformly weighted mixture of two SBNs can easily approximate p exactly by letting $q_1(\tau_1) = 1$ and $q_2(\tau_2) = 1$, resulting in $\frac{1}{2}q_1(\tau_1) + 0 = 0 + \frac{1}{2}q_2(\tau_2) = 0.5$. However, as a single SBN, q, samples A or A' and B or B' independently, and in order to achieve $q(\tau_1), q(\tau_2) > 0$, it will have to assign a non-zero probability to all four subtrees. Specifically, say $q(A) = \alpha$ and, consequently, $q(A') = 1 - \alpha$, while $q(B) = \beta$ and, consequently, $q(B') = 1 - \beta$. It follows that $q(\tau_3) = \alpha(1 - \beta)$ and $q(\tau_4) = (1 - \alpha)\beta$. Finally, $\alpha\beta = q(\tau_1) = q(\tau_2) = (1 - \alpha)(1 - \beta)$ implies $1 = \alpha + \beta$, which, in turn, implies that $q(\tau_3) = \alpha^2$ and $q(\tau_4) = \beta^2$. That is, all four trees will either have probability $1/4$ under q, or one of $\tau_3$ and $\tau_4$ will have a higher probability than $\tau_1$ and $\tau_2$. So, a single SBN can only yield a distribution that is very different from p.*

The above example exemplifies that there are tree-topology distributions that cannot be modeled using a single SBN, but can be modeled by a mixture of SBNs. There is no converse example, as a mixture can trivially model a single SBN by letting $q_1(\tau) = q_2(\tau)$ for all $\tau$. In Appendix B we construct an example that shows that these conflicting tree-topology posteriors can indeed occur for real DNA data.

## 3.1 VIMCO for Mixtures of Tree-Topology Approximations

Here we derive the VIMCO estimator of Eq. (6). Although the notation in this section is specific to Bayesian phylogenetics, our result is applicable to any mixture approximation. We purposefully follow the derivations in Mnih & Rezende (2016) closely.

### 3.1.1 Gradient Analysis

The gradients of Eq. (6) are studied first. Let

$$f(x, \mathcal{B}_s^k, \tau_s^k) = \frac{p(\mathcal{B}_s^k, \tau_s^k, X)}{\frac{1}{S}\sum_{j=1}^{S} q_{\psi_j}(\mathcal{B}_s^k|\tau_s^k)q_{\phi_j}(\tau_s^k)}, \tag{7}$$

and $\hat{L}_s^K = \log \frac{1}{K}\sum_{k=1}^{K} f(x, \mathcal{B}_s^k, \tau_s^k)$, where $\mathcal{B}_s^k, \tau_s^k$ are simulated from $q_{\psi_s, \phi_s}(\mathcal{B}, \tau)$. Note that $f(x, \mathcal{B}_s^k, \tau_s^k)$, and so also $\hat{L}_s^K$, is a function of the SBN parameters for all mixture components, i.e., $\phi_1, ..., \phi_S$. However, we omit these as arguments to the function in order to avoid cluttered notation. Furthermore, the sets of SBN parameters are disjoint (i.e. the mixture components do not share weights).

We are interested in the gradient of Eq. (6) with respect to the SBN parameters for one of the mixture components, say $i$,

$$\nabla_{\phi_i}\mathcal{L}(X; K, S) = \nabla_{\phi_i}\frac{1}{S}\sum_{s=1}^{S}\mathbb{E}_{q_{\psi_s,\phi_s}(\mathcal{B},\tau)}\left[\hat{L}_s^K\right]. \tag{8}$$

The full derivations are given in Appendix C and lead to the following expression

$$\nabla_{\phi_i}\mathcal{L}(X; K, S) = \frac{1}{S}\mathbb{E}_{q_{\psi_i,\phi_i}(\mathcal{B},\tau)}\left[\hat{L}_i^K\sum_{k=1}^{K}\nabla_{\phi_i}\log q_{\phi_i}(\tau_i^k)\right] - \\ \frac{1}{S}\sum_{s=1}^{S}\mathbb{E}_{q_{\psi_s,\phi_s}(\mathcal{B},\tau)}\left[\sum_{k=1}^{K}\tilde{w}_s^k\nabla_{\phi_i}\log\frac{1}{S}\sum_{j=1}^{S}q_{\psi_j}(\mathcal{B}_s^k|\tau_s^k)q_{\phi_j}(\tau_s^k)\right]. \tag{9}$$

We make three important observations, ($i$) for $S = 1$, we retrieve the gradients used to train VBPI, ($ii$) as the second term is negated, maximizing it promotes component $i \neq s$ to be dissimilar from component $s$, diversifying the mixture. Finally, ($iii$) the first term is merely a scaled (by $1/S$) version of the corresponding term in the $S = 1$ case. Connecting to observation ($iii$), we conclude that extending the VIMCO estimator to $S > 1$ cannot be trivially achieved without our derivation provided above.

Furthermore, the analysis of the gradients of the importance weighted lower bound in (Mnih & Rezende, 2016)—using our notation, $\mathcal{L}(X; K)$—applies here, too. That is, the gradients in the second term in Eq. (9) are multiplied by normalized weights, ensuring that the norm of the weighted sum over all $K$ gradients is not greater than the norm of the largest term in the sum. This means that $\phi_i$ will be updated mainly according to gradients based on simulations scored highly by $f$, while mitigating the impact of gradients from lower-scoring simulations.

In the first term, on the other hand, all gradients are multiplied by the same $\hat{L}_i^K$, indicating that the gradients of high-scoring simulations will not receive more weight than low-scoring ones, causing high variance and slow learning.

### 3.1.2 The VIMCO Estimator

As concluded above, the second term in Eq. (9) is well-behaved, and we do not need variance-reduction techniques to use it for learning in practice. The first term, however, requires attention in order to facilitate efficient learning.

Fortunately, as the first term is merely a scaled version of the corresponding term in the VIMCO estimator when $S = 1$, we can directly apply the localized learning signal strategy from Mnih & Rezende (2016) to obtain the VIMCO estimator for $S \geq 1$,

$$\nabla_{\phi_i}\mathcal{L}(X; K, S) \simeq \frac{1}{S}\sum_{k=1}^{K}\hat{L}_{i(k|-k)}^K\nabla_{\phi_i}\log q_{\phi_i}(\tau_i^k) - \\ \frac{1}{S}\sum_{s=1}^{S}\sum_{k=1}^{K}\tilde{w}_s^k\nabla_{\phi_i}\log\frac{1}{S}\sum_{j=1}^{S}q_{\psi_j}(\mathcal{B}_s^k|\tau_s^k)q_{\phi_j}(\tau_s^k), \tag{10}$$

where $\tau_i^k, \mathcal{B}_i^k \sim q_{\phi_i,\psi_i}(\tau, \mathcal{B})$ and $\tau_s^k, \mathcal{B}_s^k \sim q_{\phi_s,\psi_s}(\tau, \mathcal{B})$. Here, $\hat{L}_{i(k|-k)}^K$ is the local learning signal for sample $k$, defined as

$$\hat{L}_{i(k|-k)}^K = \hat{L}_i^K - \log\frac{1}{K}\left(\sum_{k'\neq k}f\left(x, \tau_i^{k'}, \mathcal{B}_i^{k'}\right) + \hat{f}\left(x, \tau_i^{-k}, \mathcal{B}_i^{-k}\right)\right), \tag{11}$$

with $\hat{f}\left(x, \tau_i^{-k}, \mathcal{B}_i^{-k}\right)$ being an estimator of $f\left(x, \tau_i^k, \mathcal{B}_i^k\right)$, typically the geometric mean Mnih & Rezende (2016); Zhang & Matsen IV (2019); Zhang (2020), $\hat{f}\left(x, \tau_i^{-k}, \mathcal{B}_i^{-k}\right) = \exp\left(\frac{1}{K-1}\sum_{k'\neq k}\log f\left(x, \tau_i^{k'}, \mathcal{B}_i^{k'}\right)\right)$.

In Appendix D we show that there is no bias introduced to the VIMCO estimator by letting the variational distribution be a uniform mixture distribution.

## 4 Experiments

In Sec. 3, we argued that a single-component approximation will struggle to properly model all parts of the target distribution when learned with black-box VI. Below, in Sec. 4.1, we experimentally verify this claim, and, furthermore, confirm that mixture components collaborate in order to jointly cover the target density, resulting in more accurate approximations and efficient exploration.

We then, in Sec. 4.2, demonstrate that the increased model flexibility and promotion of exploration translates into lower-variance estimators of the marginal log-likelihood and smaller KL divergence to an MCMC-based approximation of the true tree-topology posterior. We also visualize representations of VBPI-Mixtures on real data. Code for all experiments is provided at Github.

### 4.1 Exploring a Discrete Two-Level Hierarchical Model using Black-Box VI

Here, we examine how mixtures explore a discrete hierarchical target distribution when learned via black-box VI. We construct this experiment to analyze the capacities of discrete mixture models versus non-mixture models when the variational family of each component does not contain the target distribution. The experiment is thus aligned with Example 3.1.

Using a two-leveled hierarchical model of categorical distributions as the target distribution, $p(z_1, z_2) = p(z_2|z_1)p(z_1)$, we infer the parameters of the variational approximations by minimizing $\mathrm{KL}\left(\frac{1}{S}\sum_{s=1}^{S} q_{\phi_s}(z_1, z_2)\middle\|p(z_1, z_2)\right)$, i.e. the reverse KL. The CPD $p(z_2|z_1)$ is a categorical distribution with $n_2$ categories, conditioned on the sampled category in the previous level, $z_1$, and $p(z_1)$ is a categorical distribution with $n_1$ categories. The family of variational approximations assumes independence between $z_1$ and $z_2$, such that the $s$-th component is defined as $q_{\phi_s}(z_1, z_2) = q_{\phi_s^2}(z_2)q_{\phi_s^1}(z_1)$, where $\phi_s^2$ are the learnable category probabilities over the $n_2$ categories of component $s$. Hence a single component is insufficient to capture the posterior accurately. Meanwhile, a mixture of these approximations will be flexible enough given a sufficiently large $S$.

The parameters of $p$ are drawn from a Dirichlet distribution with all concentration parameters equal to 0.5, and the learning rates when inferring variational distributions are chosen based on a grid search on a separate target distribution. All $\phi_s$ are initialized uniformly over the categories. We compare an $S = 10$ mixture model which uses $Z = 1$ Monte Carlo estimates to estimate the objective function, with an $S = 1$ and $Z = 10$ model. This is to equate the number of samples used during learning.

We learn the approximations using either $K = 1$ or $K = 10$ importance samples. That is, when $K = 1$ the distributions are inferred by minimizing the reverse KL, and when $K = 10$ they are inferred by minimizing the importance-weighted reverse KL. The accuracies of the distributions inferred with $K = 1$ are measured in terms of the forward KL (see Fig. 3). As an additional metric of accuracy, the distributions inferred using $K = 10$ are evaluated also based on the variance of their importance weights (shown in Fig. 4b). That is, we sample 10000 importance weights, $w_i = p(z_1^i, z_2^i)/\frac{1}{S}\sum_j q_{\phi_j}(z_1^i, z_2^i)$, where $(z_1^i, z_2^i) \sim \frac{1}{S}\sum_s q_{\phi_s}(z_1^i, z_2^i)$, and compute the variance of these weights. Note that the total number of samples used is the same for all models (if $S = 1$ then $Z = 10$, and vice versa). The variance metric assesses the aptness of employing the distributions as proposal distributions (lower is better).

Regardless of choice of learning rate, the non-mixture models are not able to accurately capture the posterior. The mixture models can, on the other hand, accurately approximate the target distribution. In figures 3 and 4a-4b, we report the mentioned metrics over iterations when $n_1 = 5$ and $n_2 = 10$. All models are trained for 60k epochs, with decaying learning rates by 0.9 every 10k epochs.

### 4.2 Posterior Approximations using Real Data

We performed experiments on eight datasets (Hedges et al., 1990; Garey et al., 1996; Yang & Yoder, 2003; Henk et al., 2003; Lakner et al., 2008; Zhang & Blackwell, 2001; Yoder & Yang, 2004; Rossman et al., 2001) which we will refer to as DS1-8. These are popular datasets for evaluating Bayesian phylogenetics methods, and, as in Zhang & Matsen IV (2019); Zhang (2020); Moretti et al. (2021); Koptagel et al. (2022);

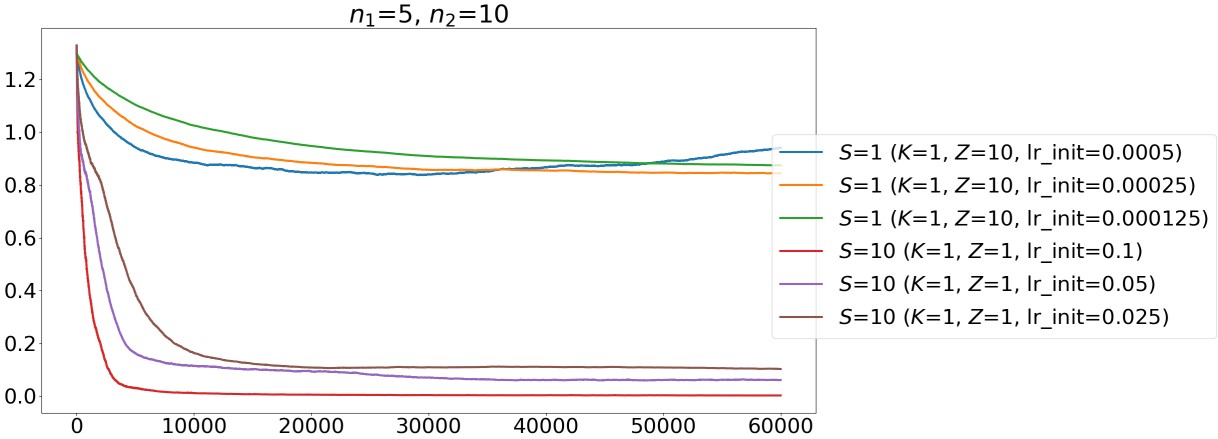

Figure 3: Forward KL divergences from the target visualized defined in Sec. 4.1 to the mixture approximations with $S = 10$ and $Z = 1$ Monte Carlo samples to estimate the objective function, and to single component approximations with $Z = 10$. On the $x$-axis are the number of training iterations. When using higher initial learning rates than those shown in the plot, the KL-curves of the single-component approximations diverged.

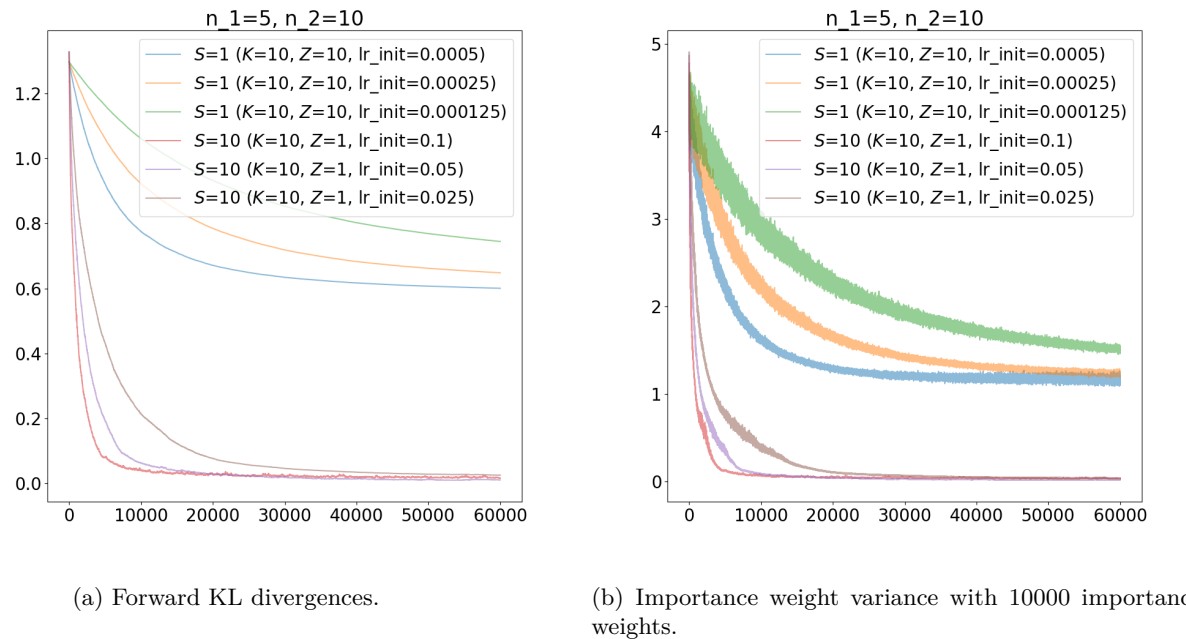

(a) Forward KL divergences.

(b) Importance weight variance with 10000 importance weights.

Figure 4: Evaluations of the accuracies of the approximations with respect to the target distribution described in Sec. 4.1.

Zhang (2023); Zhang & Matsen IV (2024), we focus on learning the approximations of branch-length and tree-topology distributions. Following the referenced works, we assume the exponential branch-length prior has rate 10 and a uniform prior over all unrooted trees (see Sec. 2 for details about the generative model). The substitution model is the Jukes-Cantor 69 model (Jukes et al., 1969), and the candidate trees, $\mathcal{T}$, are gathered from ten replicates of 10000 ultrafast maximum likelihood bootstrap trees (Minh et al., 2013). The

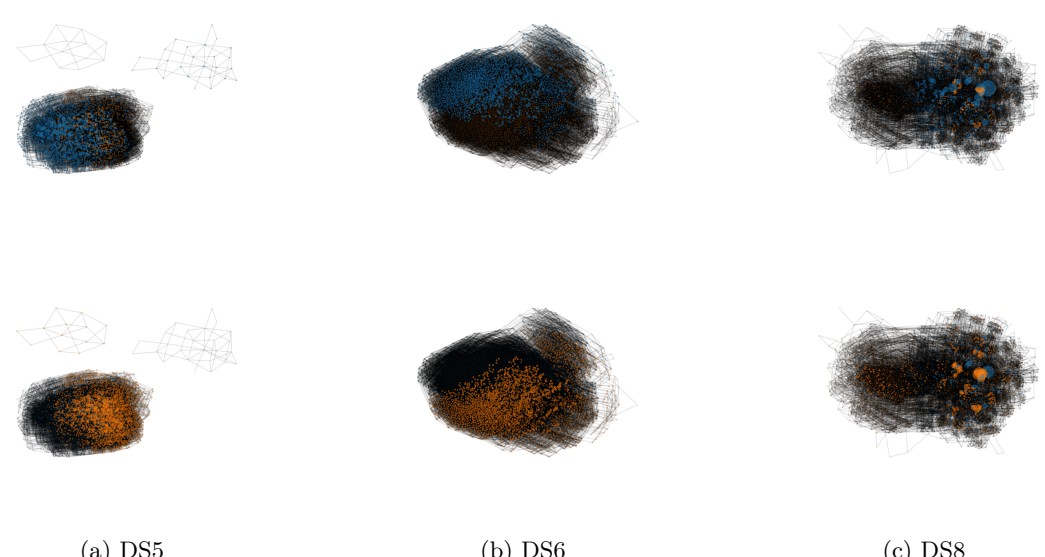

(a) DS5                     (b) DS6                     (c) DS8

Figure 5: Visualization of a uniformly weighted $S = 2$-component mixture of SBNs on (a) DS5, (b) DS6 and (c) DS8, where each node corresponds to a unique tree-topology. The upper row shows the distribution of five million sampled tree topologies from the first component, where a node, $\tau$, is colored blue if $q_{\phi_1}(\tau) > q_{\phi_2}(\tau)$, or orange otherwise. Vice versa for the lower row. The size of a node is determined by its sampling frequency, which is why nodes with low frequency appear black. The components clearly spread out, exploring different parts of the space.

implementation is based on the code provided by Zhang & Matsen IV (2024), and we trained all VBPI models during 400,000 iterations, using the same hyperparameter settings as Zhang & Matsen IV (2019); Zhang (2020). Based on the study in Zhang & Matsen IV (2024), we let $K = 10$ during training. Additionally, all SBN-based models use the same pre-computed topology sets, $\mathcal{T}$.

### 4.2.1 Visualizing the Explorative Behaviour of Mixtures of SBNs

To graphically confirm the power of employing mixtures of SBNs, we in Fig. 5 visualize representations of the learned tree-topology posterior approximations for a subset of the different real datasets. The subset was selected based on the datasets where the explorative behavior of the approximations is most clear. The representations for the other datasets, along with more implementation details, are included in the Appendix E.2. The SBNs correspond to VBPI-Mixtures without NFs. The components (upper vs. lower row) have jointly explored the space, partly specializing on disjoint sets of tree-topologies, verifying that the MISELBO objective promotes coordinated exploration of the discrete latent space, as discussed in Sec. 3.

### 4.2.2 Quantitative Evaluation of the Approximations

We quantitatively evaluate the approximations by, first, computing their KL divergences to the true posterior, and by, secondly, benchmarking their marginal log-likelihood estimates. The results are averaged over five independently trained models with different parameter initializations. All MrBayes (Huelsenbeck & Ronquist, 2001) results were produced using ten million long MCMC runs with four chains, sampling every 100 iterations. Our methods are denoted $\text{Mix}_S$, $\text{Mix}_{\text{NF},S}$, representing VBPI-Mixtures with PSPs or NFs, respectively. Mixtures that employ NFs share flow models, as described in Kviman et al. (2023).

**Statistical distances to the tree-topology posterior** Here, we compare statistical distances to the tree-topology posterior obtained from MrBayes as described above. In Table 1, the KL divergence from the posterior to the approximations, i.e., $\text{KL}(p(\tau|X)\|q_\phi(\tau))$, is computed, where $q_\phi$ represents a mixture of

Table 1: Illustrating the impact of VBPI-Mixtures in terms of $KL(p(\tau|X)\|q_\phi(\tau))$. Lower is better.

| | DS1 | DS2 | DS3 | DS4 | DS5 | DS6 | DS7 | DS8 |
|---|---|---|---|---|---|---|---|---|
| VBPI-NF | 0.0726 | 0.0110 | 0.0540 | 0.2093 | 2.2117 | 1.2842 | 0.2544 | 0.6018 |
| $\text{Mix}_{\text{NF},S=2}$ | 0.0631 | 0.0059 | 0.0475 | 0.0965 | 2.0337 | 1.0883 | 0.1183 | 0.5199 |
| $\text{Mix}_{\text{NF},S=3}$ | **0.0598** | **0.0051** | **0.0377** | **0.0769** | **1.9526** | **1.0461** | **0.0847** | **0.4567** |

Table 2: Negative marginal log-likelihood estimates on DS1-8. All VBPI methods use 1000 importance samples, and the results are averaged over 100 runs and three independently trained models. All models are trained with one Monte Carlo sample ($Z$) and ten importance samples ($K$), except for those models with names notated with specific $Z$- or $K$-values. Following Zhang & Matsen IV (2019); Zhang (2020; 2023); Zhang & Matsen IV (2024), we bold font the results with the lowest standard deviations (shown in parentheses). Details in Sec. 4.2.2. Mixtures monotonically improve with $S$.

| Data | DS1 | DS2 | DS3 | DS4 | DS5 | DS6 | DS7 | DS8 |
|---|---|---|---|---|---|---|---|---|
| # Taxa | 27 | 29 | 36 | 41 | 50 | 50 | 59 | 64 |
| # Sites | 1949 | 2520 | 1812 | 1137 | 378 | 1133 | 1824 | 1008 |
| **VBPI with Mixtures** | | | | | | | | |
| VBPI | 7108.50(0.23) | 26367.70(0.09) | 33735.10(0.14) | 13330.03(0.23) | 8214.80(0.50) | 6724.59(0.53) | 37332.12(0.45) | 8652.39(0.71) |
| $\text{VBPI}_{K=30}$ | 7123.98(0.19) | 26367.71(0.09) | 33735.09(0.08) | 13330.07(0.22) | 8215.01(0.59) | 6729.16(0.60) | 37334.43(0.46) | 8654.73(0.67) |
| $\text{VBPI}_{Z=3}$ | 7108.92(0.20) | 26367.70(0.11) | 33735.09(0.11) | 13330.06(0.24) | 8214.72(0.45) | 6725.27(0.50) | 37332.15(0.39) | 8653.40(0.52) |
| $\text{Mix}_{S=2}$ | 7108.44(0.12) | 26367.71(0.06) | 33735.10(0.07) | 13330.00(0.17) | 8214.75(0.36) | 6724.54(0.31) | 37332.04(0.24) | 8651.68(0.49) |
| $\text{Mix}_{S=3}$ | 7108.42(0.11) | 26367.71(0.04) | 33735.10(0.06) | 13329.97(0.17) | 8214.73(0.26) | 6724.51(0.28) | 37332.03(0.18) | 8650.83(0.46) |
| **VBPI with NFs and Mixtures** | | | | | | | | |
| $\text{VBPI}_{\text{NF}}$ | 7108.42(0.15) | 26367.72(0.06) | 33735.10(0.07) | 13330.00(0.23) | 8214.70(0.47) | 6724.50(0.45) | 37332.01(0.27) | 8650.68(0.46) |
| $\text{VBPI}_{\text{NF},K=30}$ | 7108.37(0.16) | 26367.70(0.05) | 33735.10(0.06) | 13329.90(0.19) | 8214.63(0.30) | 6724.38(0.32) | 37331.92(0.26) | 8650.54(0.52) |
| $\text{VBPI}_{\text{NF},Z=3}$ | 7108.37(0.17) | 26367.70(0.05) | **33735.09(0.04)** | 13329.94(0.11) | 8214.55(0.37) | 6724.37(0.39) | 37331.93(0.18) | 8650.51(0.40) |
| $\text{Mix}_{\text{NF},S=2}$ | 7108.40(0.10) | 26367.71(0.04) | 33735.10(0.05) | 13329.95(0.15) | 8214.62(0.26) | 6724.44(0.32) | 37331.96(0.19) | 8650.56(0.33) |
| $\text{Mix}_{\text{NF},S=3}$ | **7108.40(0.06)** | **26367.70(0.03)** | 33735.09(0.04) | 13329.94(0.11) | 8214.56(0.22) | **6724.40(0.23)** | **37331.96(0.15)** | 8650.54(0.30) |
| **Scores from Zhang & Matsen IV (2019), Zhang (2023), Xie & Zhang (2023), Zhou et al. (2024)** | | | | | | | | |
| $\text{MrBayes}_{ss}$ | 7108.42(0.18) | 26367.57(0.48) | 33735.44(0.50) | 13330.06(0.54) | 8214.51(0.28) | 6724.07(0.86) | 37332.76(2.42) | 8649.88(1.75) |
| GGNN | 7108.40(0.19) | 26367.73(0.10) | 33735.11(0.09) | 13329.95(0.19) | 8214.67(0.36) | 6724.38(0.42) | 37332.03(0.30) | 8650.68(0.48) |
| EDGE | 7108.41(0.14) | 26367.73(0.07) | 33735.12(0.09) | 13329.94(0.19) | 8214.64(0.38) | 6724.37(0.40) | 37332.04(0.26) | 8650.65(0.45) |
| ARTree | 7108.41(0.19) | 26367.71(0.07) | 33735.09(0.09) | 13329.94(0.17) | 8214.59(0.34) | 6724.37(0.46) | 37331.95(0.27) | 8650.61(0.48) |
| PhyloGFN | **7108.95(0.06)** | 26368.90(0.28) | 33735.60(0.35) | 13331.83(0.19) | **8215.15(0.20)** | 6739.68(0.54) | 37359.96(1.14) | **8654.76(0.19)** |

SBNs, or a single SBN, from VBPI-NF. Lower is better, and, notably, VBPI-Mixtures consistently produce KL divergences across all datasets that monotonically decrease with $S$.

**Marginal log-likelihood estimates** In terms of marginal log-likelihood estimates, we benchmark our methods against the existing VBPI algorithms: VBPI with PSP parameterization (Zhang & Matsen IV, 2019), VBPI-NF (Zhang, 2020) with ten RealNVPs (Dinh et al., 2016), and VBPI-GNN (Zhang (2023); EDGE and GGNN). Additionally, we compare our results with the stepping-stone (SS; Xie et al. (2011)) method applied to MrBayes as well as to PhyloGFN (Zhou et al., 2024) and ARTree (Xie & Zhang, 2023). The results are given in Table 2. All models have been trained using a single Monte Carlo sample and ten importance samples except for those marked with $Z$ and $K$ respectively. Following Zhang & Matsen IV (2019); Zhang (2020; 2023); Zhang & Matsen IV (2024), we bold font the results with the lowest standard deviations. Rewarding low-variance estimates is motivated, as they imply, for instance, more reliable Bayesian model selections for downstream tasks. We can see that the PhyloGFN has achieved good performance regarding standard deviation on three of the datasets. However, PhyloGFN does seem to struggle with producing competitive mean marginal log-likelihood. For VBPI-Mixture, increasing the number of mixture components results in significant improvements in terms of lower standard deviations (on all datasets) and higher mean marginal log-likelihood scores (especially apparent on the more complex datasets, e.g. DS5-8).

**Trade-off between marginal log-likelihood and standard deviations** As mentioned above, in the VBPI literature, the target metric in the marginal log-likelihood estimation task is frequently considered to be the standard deviation. I.e., the lower the standard deviation, the better, with less focus on the mean marginal log-likelihood results. The reason for this might be that the mean marginal log-likelihood scores on DS1-8 are quite saturated, which naturally instead shifts the focus to the standard deviations of the estimates. In Xie & Zhang (2023), they alternatively motive the standard deviation criterion by pointing

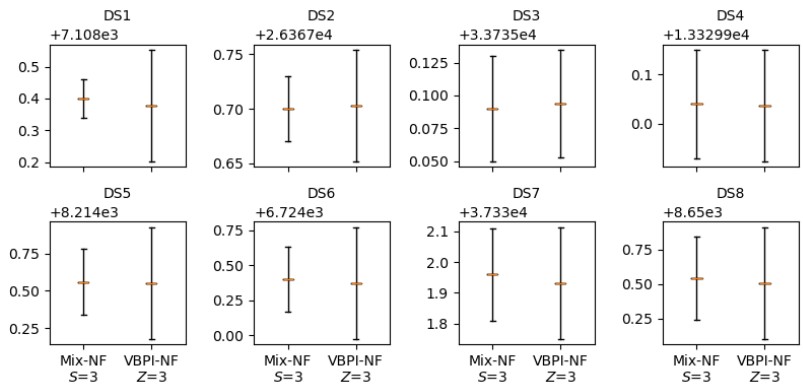

Figure 6: Illustrations of the mean and standard deviations of the (negative) marginal log-likelihood scores from two of the models. The plots are complementary to the scores in Table 2, critically demonstrating cases where VBPI with $Z = 3$ can achieve lower mean scores, while our VBPI-Mixture with $S = 3$ consistently produces smaller or similar standard deviations.

to the importance sampling domain, stating that the variance of the estimator is a good reflection of the accuracy between the importance sampling distribution and the target distribution.

Nonetheless, as we observe that the mean log-likelihoods can be slightly improved when using $Z = 3$ Monte Carlo samples, we additionally plot the mean log-likelihood vs. standard deviation trade-off between our $S = 3$ VBPI-NF-Mixture and the $Z = 3$ VBPI-NF. This provides the practitioner with more information when deciding which method to employ. Please see Fig. 6.

**Compute infrastructure and run times**  Most computations have been conducted on an AMD EPYC 7742 where two cores have been used per run. Final runtimes are shown in Table 3. The table shows the joint run time for both training and testing. Also worth noting is that multiple mixture components also multiply the number of particles, so the majority of the time increase is due to the Felsenstein pruning algorithm for the likelihood model evaluation, which grows linearly.

Finally, and crucially, the code used was not optimized for run time, and so a wall-clock time is not an apt metric for comparisons.

Table 3: Compute time reported in minutes for fitting the model as well as evaluating the marginal likelihood and continuous estimate of ELBO while training every 5000 iterations using 1000 samples with a single particle. The Ufboot2 was run on i9-13900k, the command used for a single repetition for DS1 `./iqtree2 -s DS1.fasta -bb 10000 -wbt -m JC69`.

| Data | DS1 | DS2 | DS3 | DS4 | DS5 | DS6 | DS7 | DS8 |
|---|---|---|---|---|---|---|---|---|
| VBPI | 735.4 | 813.95 | 962.67 | 1087.98 | 1273.57 | 1299.55 | 1606.03 | 1660.18 |
| $Mix_{S=2}$ | 1984.47 | 2172.18 | 2598.17 | 2888.58 | 3322.77 | 3389.5 | 4127.52 | 4234.98 |
| $Mix_{S=3}$ | 3224.57 | 3425.35 | 4056.07 | 4493.87 | 5614.58 | 5604.43 | 6705.18 | 7004.28 |
| VBPI-NF | 900.32 | 953.8 | 1150.78 | 1278.27 | 1585.87 | 1523.02 | 1869.23 | 1974.17 |
| $Mix_{NF,S=2}$ | 2385.38 | 2442.75 | 2893.15 | 3300.35 | 3800.88 | 3728.75 | 4647.48 | 4754.22 |
| $Mix_{NF,S=3}$ | 3669.8 | 3939.73 | 4576.08 | 5180.75 | 6206.52 | 6220.4 | 7568.85 | 7776.37 |
| Ufboot2 | 0.19 | 0.28 | 0.27 | 0.18 | 0.11 | 0.15 | 0.46 | 0.13 |

## 5 Conclusion

We introduced VBPI-Mixtures, a novel algorithm that increases the flexibility of the phylogenetic posterior approximation by utilizing recent advances in mixtures for black-box VI. We showed that mixtures of SBNs

can approximate distributions that a single SBN cannot, making a persuasive case for VBPI-Mixtures. Experimentally, we achieve state-of-the-art results on eight popular real phylogenetics datasets in terms of producing lower-variance estimators of the marginal log-likelihood, show that mixtures of SBNs can accurately capture tree-topology posteriors that non-mixture SBNs cannot, and produce KL divergences to an MCMC-based approximation of the true tree-topology posterior that decrease with $S$.

The drawbacks of this framework primarily lie in the increased computational time 4.2.2 compared to the baselines and the dependency on pre-computed topologies, both of which create a higher barrier to entry for downstream tasks. However, we are confident that the compute time can be significantly reduced with a greater focus on parallelization.

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

## A  A Brief Introduction to Bayesian Phylogenetic Inference for Machine Learning Researchers

This introduction aims to briefly explain the basic concepts required to understand the generative model provided in the main text.

Phylogenetic trees capture evolutionary relationships among species and provides valuable insights into life's evolutionary history. Within this domain, phylogenies are often depicted as bifurcating tree graphs, where nodes represent common ancestors, and branches (edges) signify evolutionary events and genetic distances between species. This framework enables an understanding of species relatedness, ancestry, and the evolutionary processes governing life's diversity.

Bayesian phylogenetic inference builds upon this framework by applying Bayesian statistical methods to infer the evolutionary history. It allows for a probabilistic approach to model uncertainty and variation, considering prior beliefs about evolutionary parameters and updating these beliefs as new data is incorporated. It is common to use DNA or protein sequences as data since it describes different attributes of the species, and the edges represent a mutation between species. Through sampling from a posterior distribution and utilizing tools like Markov Chain Monte Carlo (MCMC) methods or variational inference, Bayesian phylogenetics offers a robust and nuanced view of evolutionary relationships, integrating multiple sources of information and providing a rigorous statistical foundation for evolutionary hypotheses.

Mainly, two latent variables are regarded as important in Bayesian phylogenetic inference. First, the tree topology, $\tau$, a binary tree with the observations assigned to its leaves. The tree-topology space grows as $(2n-3)!!$ for rooted and $(2n-5)!!$ for unrooted trees, where $n$ is the number of leaves (observations/taxa). Furthermore, each edge, $e$, of the topology is associated with a positive continuous variable, the branch lengths, $b(e)$. The Cartesian product of discrete and continuous spaces makes inference in phylogenetics a challenging task.

## B  Conflicting Tree Topologies

Here we construct a realistic scenario where DNA sequences induce conflicting tree-topologies in the posterior that cannot be modelled by the vanilla SBN. On the other hand, they can be captured by VBPI-Mixtures.

It is well-known that DNA data sometimes has conflicting signals. Here we construct a toy example to demonstrate how $\tau_1$ and $\tau_2$ can have higher posterior support than $\tau_3$ and $\tau_4$. For simplicity, we use the connection between a lower parsimony score and a higher likelihood when branch lengths are short (to appear in the appendix). First, consider that the leaves in Fig. 2 have DNA sequences with nucleotides at sites $i$ and $j$ specified in the table below (the first and second columns represent $i$ and $j$, respectively).

Table 4: Nucleotide assignments to sites $i$ and $j$ in the observations.

| Sites | (1) | (2) | (3) | (4) | (5) | (6) |
|---|---|---|---|---|---|---|
| $i$ | A | C | C | A | G | G |
| $j$ | C | C | A | G | G | A |

We compute the parsimony scores for each clade (A, A', B, B') and the cost of joining two clades to form $\tau_1, ..., \tau_4$. The cost of transitioning from one nucleotide (A, C, G, T) to another is 1.

In the first table below, we start by calculating all possible parsimony scores for each clade. Here X→Y denotes the parsimony score, Y, if the nucleotide at site $i$ or $j$ in the ancestral sequence in the root is X. Bolded is the best (lowest) score.

Table 5: Parsimony scores for the four different subtrees

| MP | $A = ((1,2),3)$ | $B = ((4,5),6)$ | $A' = (1,(2,3))$ | $B' = (4,(5,6))$ |
|---|---|---|---|---|
| $i$:th | A→2,**C→1**,G→3,T→3 | A→2,C→3,**G→1**,T→3 | **A→1**,**C→1**,G→2,T→2 | **A→1**,C→2,**G→1**,T→2 |
| $j$:th | **A→1**,**C→1**,G→2,T→2 | **A→1**,C→3,**G→1**,T→3 | A→2,**C→1**,G→3,T→3 | A→2,C→3,**G→1**,T→3 |

Table 6: Parsimony scores for all possible tree topologies

| MP | $\tau_1(A \wedge B)$ | $\tau_2(A' \wedge B')$ | $\tau_3(A \wedge B')$ | $\tau_4(A' \wedge B)$ |
|---|---|---|---|---|
| $i$:th | A→4,**C→3**,**G→3**,T→4 | **A→2**,C→3,G→3,T→4 | **A→3**,**C→3**,**G→3**,T→4 | **A→3**,**C→3**,**G→3**,T→4 |
| $j$:th | **A→2**,C→3,G→3,T→4 | A→4,**C→3**,**G→3**,T→4 | **A→3**,**C→3**,**G→3**,T→4 | **A→3**,**C→3**,**G→3**,T→4 |

Now we can see that $\tau_1$ and $\tau_2$ give us a better parsimony score, demonstrating that the scenario exemplified in Fig. 2 can occur in biological data when there are conflicting signals.

We direct readers who want to learn more about parsimony scores to Chapter 1 in Felsenstein (2003).

## C  Gradient Derivation

Here we show the full derivations of the gradients w.r.t. $\phi_i$. For completeness, recall that

$$f(x, \mathcal{B}_s^k, \tau_s^k) = \frac{p(\mathcal{B}_s^k, \tau_s^k, X)}{\frac{1}{S} \sum_{j=1}^{S} q_{\psi_j}(\mathcal{B}_s^k | \tau_s^k) q_{\phi_j}(\tau_s^k)}, \tag{12}$$

and $\hat{L}_s^K = \log \frac{1}{K} \sum_{k=1}^{K} f(x, \mathcal{B}_s^k, \tau_s^k)$, where $\mathcal{B}_s^k, \tau_s^k$ are simulated from $q_{\psi_s, \phi_s}(\mathcal{B}, \tau)$.

That is, we are interested in the gradient of Eq. (6) w.r.t. the SBN parameters for one of the mixture components, say $i$,

$$\nabla_{\phi_i} \mathcal{L}(X; K, S) = \nabla_{\phi_i} \frac{1}{S} \sum_{s=1}^{S} \mathbb{E}_{q_{\psi_s, \phi_s}(\mathcal{B}, \tau)} \left[ \hat{L}_s^K \right]. \tag{13}$$

There are two cases to take into account in the sum, either $i = s$ or $i \neq s$. Starting with $i = s$ and using the product rule,

$$\nabla_{\phi_i} \frac{1}{S} \mathbb{E}_{\mathcal{B}_i^{1:K}, \tau_i^{1:K} \sim q_{\psi_i, \phi_i}(\mathcal{B}, \tau)} \left[ \hat{L}_i^K \right] = \nabla_{\phi_i} \frac{1}{S} \sum_{\tau_i^{1:K}} q_{\phi_i}(\tau_i^{1:K}) \mathbb{E}_{\mathcal{B}_i^{1:K} \sim q_{\psi_i}(\mathcal{B}|\tau_i^{1:K})} \left[ \hat{L}_i^K \right] \tag{14}$$

$$= \frac{1}{S} \sum_{\tau_i^{1:K}} \mathbb{E}_{\mathcal{B}_i^{1:K} \sim q_{\psi_i}(\mathcal{B}|\tau_i^{1:K})} \left[ \nabla_{\phi_i} q_{\phi_i}(\tau_i^{1:K}) \hat{L}_i^K \right] \tag{15}$$

$$= \frac{1}{S} \sum_{\tau_i^{1:K}} \mathbb{E}_{q_{\psi_i}(\mathcal{B}|\tau_i^{1:K})} \left[ \hat{L}_i^K \nabla_{\phi_i} q_{\phi_i}(\tau_i^{1:K}) + q_{\phi_i}(\tau_i^{1:K}) \nabla_{\phi_i} \hat{L}_i^K \right]. \tag{16}$$

Recalling the identity that $\nabla_\phi g_\phi(z) = g_\phi(z) \nabla_\phi \log g_\phi(z)$, we start by rewriting the first term inside the expectation in Eq. (16) as

$$\hat{L}_i^K \nabla_{\phi_i} q_{\phi_i}(\tau_i^{1:K}) = q_{\phi_i}(\tau_i^{1:K}) \hat{L}_i^K \nabla_{\phi_i} \log q_{\phi_i}(\tau_i^{1:K}) \tag{17}$$

$$= q_{\phi_i}(\tau_i^{1:K}) \hat{L}_i^K \sum_{k=1}^{K} \nabla_{\phi_i} \log q_{\phi_i}(\tau_i^k), \tag{18}$$

and then the second term

$$q_{\phi_i}(\tau_i^{1:K}) \nabla_{\phi_i} \hat{L}_i^K = q_{\phi_i}(\tau_i^{1:K}) \nabla_{\phi_i} \log \frac{1}{K} \sum_{k=1}^{K} f(x, \mathcal{B}_i^k, \tau_i^k) \tag{19}$$

$$= q_{\phi_i}(\tau_i^{1:K}) \sum_{k=1}^{K} \tilde{w}_i^k \nabla_{\phi_i} \log f(x, \mathcal{B}_i^k, \tau_i^k) \tag{20}$$

$$= -q_{\phi_i}(\tau_i^{1:K}) \sum_{k=1}^{K} \tilde{w}_i^k \nabla_{\phi_i} \log \frac{1}{S} \sum_{j=1}^{S} q_{\psi_j}(\mathcal{B}_i^k|\tau_i^k) q_{\phi_j}(\tau_i^k), \tag{21}$$

where $\tilde{w}_i^k = \frac{f(x, \mathcal{B}_i^k, \tau_i^k)}{\sum_{k'=1}^{K} f(x, \mathcal{B}_i^{k'}, \tau_i^{k'})}$. Exchanging the two terms in Eq. (16) with Eq. (18) and (21), respectively, we get

$$\nabla_{\phi_i} \frac{1}{S} \mathbb{E}_{q_{\psi_i, \phi_i}(\mathcal{B}, \tau)} \left[ \hat{L}_i^K \right] = \frac{1}{S} \mathbb{E}_{q_{\psi_i, \phi_i}(\mathcal{B}, \tau)} \left[ \hat{L}_i^K \sum_{k=1}^{K} \nabla_{\phi_i} \log q_{\phi_i}(\tau_i^k) \right] - \tag{22}$$

$$\frac{1}{S} \mathbb{E}_{q_{\psi_i, \phi_i}(\mathcal{B}, \tau)} \left[ \sum_{k=1}^{K} \tilde{w}_i^k \nabla_{\phi_i} \log \frac{1}{S} \sum_{j=1}^{S} q_{\psi_j}(\mathcal{B}_i^k|\tau_i^k) q_{\phi_j}(\tau_i^k) \right]. \tag{23}$$

For $i \neq s$ we may move the gradient operator into the expectation directly and reuse the derivation of Eq. (21),

$$\nabla_{\phi_i} \frac{1}{S} \sum_{s \neq i} \mathbb{E}_{q_{\psi_s, \phi_s}(\mathcal{B}, \tau)} \left[ \hat{L}_s^K \right] = \frac{1}{S} \sum_{s \neq i} \mathbb{E}_{q_{\psi_s, \phi_s}(\mathcal{B}, \tau)} \left[ \nabla_{\phi_i} \hat{L}_s^K \right] \tag{24}$$

$$= -\frac{1}{S} \sum_{s \neq i} \mathbb{E}_{q_{\psi_s, \phi_s}(\mathcal{B}, \tau)} \left[ \sum_{k=1}^{K} \tilde{w}_s^k \nabla_{\phi_i} \log \frac{1}{S} \sum_{j=1}^{S} q_{\psi_j}(\mathcal{B}_s^k|\tau_s^k) q_{\phi_j}(\tau_s^k) \right].$$

Considering both cases, we return to Eq. (13),

$$\nabla_{\phi_i} \mathcal{L}(X; K, S) = \frac{1}{S} \mathbb{E}_{q_{\psi_i, \phi_i}(\mathcal{B}, \tau)} \left[ \hat{L}_i^K \sum_{k=1}^{K} \nabla_{\phi_i} \log q_{\phi_i}(\tau_i^k) \right] -$$

$$\frac{1}{S} \sum_{s=1}^{S} \mathbb{E}_{q_{\psi_s, \phi_s}(\mathcal{B}, \tau)} \left[ \sum_{k=1}^{K} \tilde{w}_s^k \nabla_{\phi_i} \log \frac{1}{S} \sum_{j=1}^{S} q_{\psi_j}(\mathcal{B}_s^k|\tau_s^k) q_{\phi_j}(\tau_s^k) \right], \tag{25}$$

which is the expression for the gradient we need in order to apply the VIMCO estimator (see Sec. 3.1.2).

## D  Unbiasedness of the VIMCO Estimator Applied to Uniform Mixtures

Recall that the VIMCO estimator applied to a uniformly weighted mixture of variational distributions is given in Eq. (10). We want to analyze the expectation of the R.H.S. of this equation. Note also that $\phi_1, ..., \phi_S$ are disjoint sets of parameters, and so they are updated via the gradients independently.

We now expand Eq. (10) using Eq. (11), and apply expectations:

$$\frac{1}{S}\mathbb{E}_{q_{\psi_i,\phi_i}(\mathcal{B},\tau)}\left[\sum_{k=1}^{K}\hat{L}_i^K\nabla_{\phi_i}\log q_{\phi_i}\tau_i^k)\right] \tag{26}$$

$$-\frac{1}{S}\mathbb{E}_{q_{\psi_i,\phi_i}(\mathcal{B},\tau)}\left[\sum_{k=1}^{K}\log\frac{1}{K}\left(\sum_{k'\neq k}f\left(x,\tau_i^{k'},\mathcal{B}_i^{k'}\right)+\hat{f}\left(x,\tau_i^{-k},\mathcal{B}_i^{-k}\right)\right)\nabla_{\phi_i}\log q_{\phi_i}(\tau_i^k)\right] \tag{27}$$

$$-\frac{1}{S}\sum_{s=1}^{S}\mathbb{E}_{q_{\psi_s,\phi_s}(\mathcal{B},\tau)}\left[\sum_{k=1}^{K}\tilde{w}_s^k\nabla_{\phi_i}\log\frac{1}{S}\sum_{j=1}^{S}q_{\psi_j}(\mathcal{B}_s^k|\tau_s^k)q_{\phi_j}(\tau_s^k)\right] \tag{28}$$

Now, using the REINFORCE identity, we can rewrite the first and the third term above (for more details, you can go backwards in our gradient derivations from Eq. (22)) to show that:

$$\frac{1}{S}\mathbb{E}_{q_{\psi_i,\phi_i}(\mathcal{B},\tau)}\left[\hat{L}_i^K\sum_{k=1}^{K}\nabla_{\phi_i}\log q_{\phi_i}(\tau_i^k)\right]-\frac{1}{S}\mathbb{E}_{q_{\psi_i,\phi_i}(\mathcal{B},\tau)}\left[\sum_{k=1}^{K}\tilde{w}_i^k\nabla_{\phi_i}\log\frac{1}{S}\sum_{j=1}^{S}q_{\psi_j}(\mathcal{B}_i^k|\tau_i^k)q_{\phi_j}(\tau_i^k)\right] = \tag{29}$$

$$\nabla_{\phi_i}\frac{1}{S}\mathbb{E}_{q_{\psi_i,\phi_i}(\mathcal{B},\tau)}\left[\hat{L}_i^K\right] = \nabla_{\phi_i}\mathcal{L}(X;K,S) \tag{30}$$

From this equality, and by subtracting the second term in Eq. 26 from $\nabla_{\phi_i}\mathcal{L}(X;K,S)$, we get

$$\nabla_{\phi_i}\mathcal{L}(X;K,S)-\frac{1}{S}\mathbb{E}_{q_{\psi_i,\phi_i}(\mathcal{B},\tau)}\left[\sum_{k=1}^{K}\log\frac{1}{K}\left(\sum_{k'\neq k}f\left(x,\tau_i^{k'},\mathcal{B}_i^{k'}\right)+\hat{f}\left(x,\tau_i^{-k},\mathcal{B}_i^{-k}\right)\right)\nabla_{\phi_i}\log q_{\phi_i}(\tau_i^k)\right], \tag{31}$$

where the second term is the control variate. The control variate is, however, independent of the mixture formulation (except for the $1/S$ scaling which does not affect the unbiasedness of the term). As such, since VIMCO is an unbiased estimator of the gradient (Mnih & Rezende, 2016), then so is our version of it.

## E  Additional Experimental Results and Implementation Details

### E.1  The Two-Level Hierarchical Model

Running a grid search for all five algorithms ($S = 1, ..., 5$), using $n_1 = 5$ and $n_2 = 10$, we found that the learning rates 0.01, 0.1, 0.1, 0.2, and 0.25, respectively, were optimal. That is, these achieved the smallest KL divergences. For larger learning rates, $S = 1$ did not converge or converge to worse KL divergences. The optimal learning rates found in the grid searches were used in all subsequent experiments.

In Fig. 7, we visualize the KL curves as functions of the number of training iterations. For all configurations of $K$, $n_1$ and $n_2$, the $S = 5$ model performs best. The pattern—$S = 1$ converges slower and to worse KL divergences than $S > 2$—holds also when all models use the same number of importance samples, $K$ (shown in Fig 7).

### E.2  Visualization Details and More Plots

We employed the software as in Whidden & Matsen IV (2015), i.e. rSPR was used to determine the distances between the topologies. Additionally, we adopted the same methodology for cluster creation. This involved

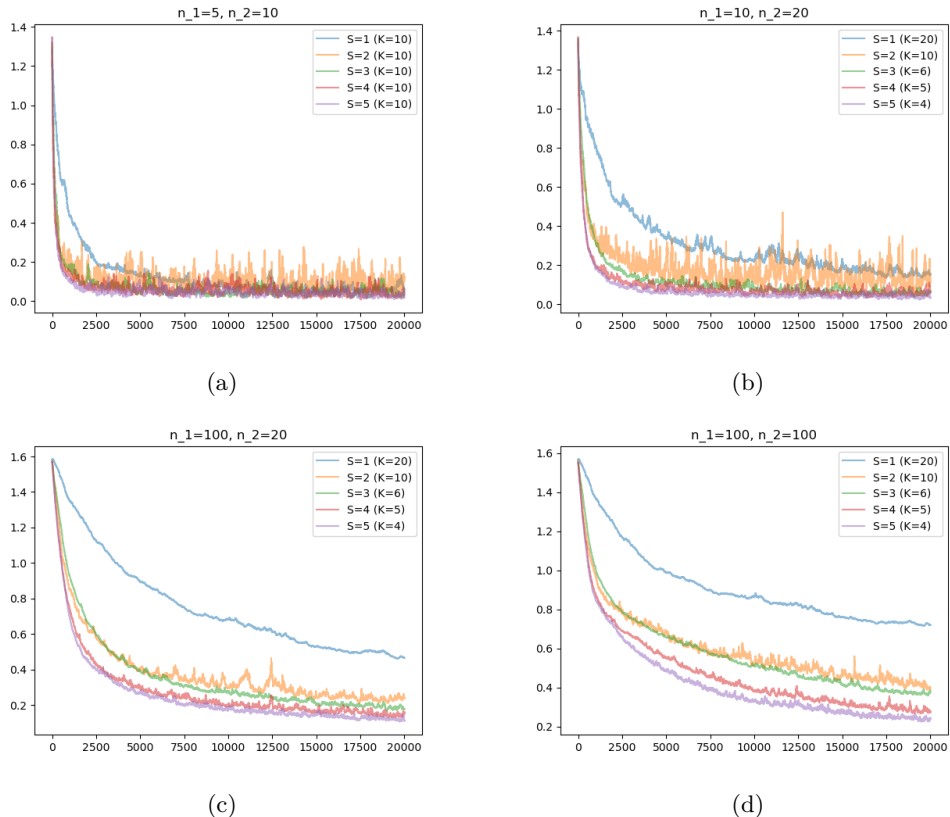

Figure 7: KL curves for the two-level hierarchical model using different configurations of $K$, $n_1$ and $n_2$.

assigning the most probable peak to a cluster and subsequently assigning all unassigned trees to the same cluster if their distance from the peak tree was within one standard deviation below the mean distance of all unassigned trees. This iterative process continued until all topologies were assigned, or until eight clusters were reached.

For graph creation, we employ the Graphviz layout known as Scalable Force-Directed Placement (SFDP), in conjunction with the NetworkX library Hagberg et al. (2008). The clusters are represented by different colors, and the size of each node is determined by the normalized sampling frequency in Fig 1, 8. Moreover, edges between the topologies are only displayed if their distance is exactly one. It's worth noting that the rSPR distance measure is utilized, which counts the number of changes similar to the approach used in the MCMC method.

To ensure that the visualization focused on the most credible information, we imposed a constraint by limiting the nodes to the 95% most credible set. This ensured that only the most reliable nodes were included. Additionally, to manage computational resources effectively, we set a maximum limit of 4096 nodes for the graphs.

A similar approach was used for Fig. 5, 9, with the main difference being that we sampled from the components and displayed the joint set of topologies. The colors in this figure were based on which component sampled the topology the most. This representation was considered an approximation of the posterior.

## F  Limitations

We use $S$ SBNs and branch length models to form our mixture approximations. This introduces a larger number of model parameters. In our current implementation, the training time was prolonged as we did

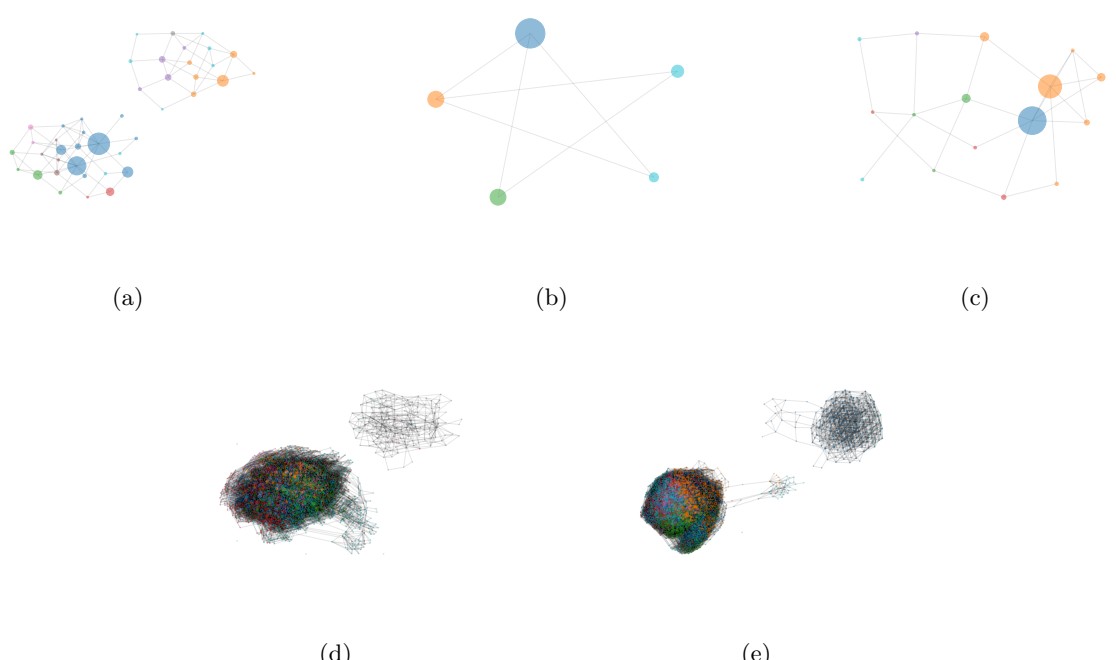

(a)  (b)  (c)

(d)  (e)

Figure 8: Visualization of samples from the tree-topology posterior using a 1,000,000,000 iterations long MCMC run on (a) DS1, (b) DS2, (c) DS3, (d) DS5 and (e) DS6. Nodes represent unique tree-topologies and are colored based on cluster assignments, illustrating the multimodality of the tree-topology posterior. More details in Sec. 2.

not parallelize the parameter updates of the parameters of the mixture components. This can, however, be done, in order to heavily decrease training time.

Additionally, using shared parameters for the mixture components can also be utilized, if the practitioner is running on a limited memory budget. However, shared weights would violate the assumption of disjoint SBN parameter sets in Appendix D. Nonetheless, devising clever modifications to reduce the number of parameters and training times for mixtures in black-box VI is an exciting future research field.

## G   Broader Impact

Bayesian phylogenetic inference algorithms are crucial for researchers to reason about uncertainty in their evolutionary findings. Variational inference algorithms provide a compelling alternative to MCMC-based algorithms as a parametric approximation is obtained. This implies that VI, and VBPI specifically, can be used in settings where the application of MCMC is less straightforward, for instance in out-of-distribution detection, or evaluation on held-out data. Also, as we have shown in our experiments in this paper, model evaluation can be more robust when using VI over MCMC, resulting in a smaller variance of the estimator of the marginal log-likelihood. This is an important feature for downstream tasks.

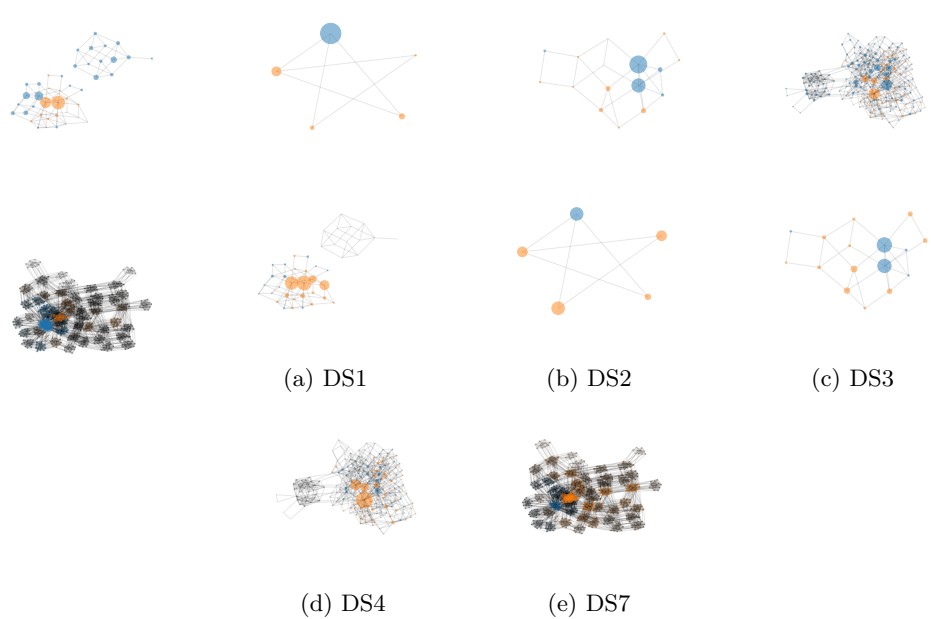

(a) DS1      (b) DS2      (c) DS3

(d) DS4      (e) DS7

Figure 9: Visualization of a uniformly weighted $S = 2$-component mixture of SBNs on (a) DS1, (b) DS2 and (c) DS3, (d) DS4 and (e) DS7, where each node corresponds to a unique tree-topology. The upper row shows the distribution of five million sampled tree topologies from the first component, where a node, $\tau$, is colored blue if $q_{\phi_1}(\tau) > q_{\phi_2}(\tau)$, or orange otherwise. Vice versa for the lower row. The size of a node is determined by its sampling frequency, which is why nodes with low frequency appear black. The components clearly spread out, exploring different parts of the space.

