# OpenReview forum: "Improved Variational Bayesian Phylogenetic Inference using Mixtures"
_TMLR — Accepted by TMLR_

### Review · Reviewer_LrA3 · 2024-10-09

**Summary Of Contributions:**

The paper describes a mixture approach for variational Bayesian phylogenetics, where the goal is to perform probabilistic inference of the posterior over evolutionary topologies. Using a simple example, the authors motivate that  existing work that uses a single subsplit Bayesian network (SBN) may not accurately approximate this posterior. To resolve this, the authors propose using mixtures of SBNs, which increases the flexibility of the variational posterior and may be able to better capture the true posterior. The authors also extend existing analytical results on discrete variational inference with importance-weighted bounds (VIMCO) to variational mixtures.

**Audience:**

Yes

**Broader Impact Concerns:**

None of note.

**Claims And Evidence:**

Yes

**Requested Changes:**

Please see the weaknesses above.

**Strengths And Weaknesses:**

## Strengths

* The paper is written clearly and the derivations seem correct.
* The case for using a mixture variational family is well-motivated via Example 3.1.
* The method is simple to implement and the authors plan to share the code to reproduce it.
* The evaluation on the toy model illustrates the advantages of mixtures well, thus supporting Example 3.1.

## Weaknesses

The authors have addressed the major weaknesses from the previous submission. Below are some less critical points that would help to improve the paper.

### Additional metrics

In Table 1 and Figure 3, the authors use forward KL divergence, $\text{KL}(p(\tau \mid X) \mid\mid q\_{\phi}(\tau))$, to measure the accuracy of the variational approximation. (Figure 3 doesn't specify which KL is used but it seems to be the forward KL in the provided code---this should also be clarified in the caption.) But, the objectives optimise an _importance-weighted_ version of a reverse KL. This means, that the variational distribution should be used as importance sampling proposal distribution, e.g. by sampling from $q\_{\phi}(\tau)$ and then resampling using the importance weights $p(\tau \mid X) / q\_{\phi}(\tau)$, instead of treating $q\_{\phi}(\tau)$ as a direct approximation of the posterior $p(\tau \mid X)$ [1]. As such, it would be useful if the accuracy of the variational distribution would be assessed using some importance sampling metrics, such as the variance of the importance weights which should be low if $q$ is a good proposal.

* In Table 1, it would also be good to include the results for $\text{VBPI}\_{Z=3}$, which seem to be very close to $\text{Mix}\_{\text{NF},S=3}$ in terms of MLL, and thus would be good to know their posterior accuracy.

References:

[1] Cremer et al (2017). Reinterpreting Importance-Weighted Autoencoders.

### Clarity

* In the introduction, the authors state that "Using a toy experiment [...], we first show that the mixture components specialize in different parts of the solution space". However, the re-worked experiment section 4.1 does not directly show this, so the claim must be adjusted.
* In Section 3, first sentence of the "Mixtures promote exploration" paragraph states: "Note that minimizing the denominator in Eq. (6) corresponds to diversifying the mixture distribution". However, due to the log-sum in the objective, minimizing _just_ the denominator does not make much sense (because it doesn't factor into an entropy term like in the standard ELBO). Hence, the statement needs to be clarified.
* Sections 3.1.1 and 3.1.2 implicitly assume that the parameters of the mixture components are disjoint, as explained in Appendix D. This assumption should also be highlighted in the main text.
	* Moreover, In Appendix F, the authors suggest that using shared parameters for the mixture components could be used to reduce memory budget. But, this would violate the assumptions before and should also be highlighted.
* The observation in Section 3.1.1 "(ii) as the second term is negated, maximizing it corresponds to diversifying the mixture distribution w.r.t. $\phi_i$," is unclear. It is not clear what it means to "diversity the mixture w.r.t. $\phi_i$", or why exactly does maximizing the term correspond to any kind of diversification. If there are prior work on this, then a precise reference might suffice.
* The second paragraph in Section 4 claims that "the increased model flexibility and promotion of exploration translates
into better marginal log-likelihood estimates". But the results mostly rely on improvements to the standard deviation of the MLL, so this claim should be adjusted.

### Minor typos:

* Page 2, first sentence of section 2. $X$ is not yet introduced, hence confusing.
* Page 2, after eq 2, "reverses"->"reverse"
* Section 4.1, last paragraph, second sentence. Two typos. "This [is] to equate ... samples use[d] during learning."
* Section 4.1. last paragraph "Figures 3" -> "Figure 3"
* Figure 7: (b) is the same as (a). Should probably be D2?
* Appendix D, third sentence "Not"->"Note"

---

> ### Author Response · Authors · 2024-10-23
> **Comment 1/2**
>
> Dear reviewer,
>
> Thank you for carefully reading our revised version, all your comments are greatly appreciated.
>
> We have submitted a revised version of our submission, incorporating the mentioned clarity concerns (see Comment 2 below) and fixing the typos. Also, we have updated the experiment in Sec. 4.1 with the requested IS variance metric (see below for details). Please let us know if any clarity issues remain given the new submission. Once again, thank you!
>
> Next we discuss the request of additional metrics.
>
> ## Additional metrics
> Starting with the comments regarding the reverse KL and the additional variance metric, we would like to make the following distinction. You are correct that Figure 3 displays the (forward) KL divergences over training iterations, however none of the distributions in *Sec. 4.1* are inferred by optimizing using the importance-weighted reverse KL. As we recall, this was actually agreed upon in the previous submission round. Futhermore, although we will make sure to note this more explicitly, we let $K=1$ (see the plot legends and the text in Sec. 4.1).
>
> Nonetheless, the distributions in Table 1 are indeed inferred using the importance weighted version of the reverse KL. Now, we want to thank you for the reference (it was a very nice read), and we agree that variance of the importance weights is a good metric for evaluating the quality of a proposal distribution, however, in all honesty, we do no longer have access to the inferred models used to generate the results in Table 1. This is due to administrative issues at our affiliation. Hence, to produce the requested metrics, we would have to do new costly training runs, further overshooting our budget for this project. The same unfortunately goes for the request to update Table 1 with $\text{VBPI}_{Z=3}$.
>
> As such, we are asking if you can consider settling for the following compromise: Instead of re-training new models and obtaining a new MCMC golden run in order to complement the results in Table 1 by quantifying the variance of the importance weights, we have augmented our experiment in Sec. 4.1 with a new set of runs where the distributions are inferred using the importance weighted reverse KL. The quality of the distributions are evaluated according to the variance of their importance weights (see Fig. 4b). In this extra experiment, all optimizers minimize an objective with $K=10$, and either $Z=10$ or $S=10$ such that the total number of samples are equal. The variance is taken over 10,000 importance weights, the same number for all inferred distributions.
>
> We hope that you will find this revision fair, kindly taking into account also the many additional experiments produced in the previous submission round. If not, please let us know!

---

> > ### Author Response · Authors · 2024-10-23
> > **Comment 2/2**
> >
> > ## Clarity updates
> > You will find these updates in the revised submission, too, but for ease of inspection, we also list our updates here.
> >
> > * "Using a toy experiment [...], we first show that the mixture components specialize in different parts of the solution space" --> "Using a toy experiment where we design a complicated hierarchical categorical target distribution and pick a variational family such that it does not contain the target distribution, we first show that the mixtures of these sub-optimal variational distributions achieve smaller Kullback-Leibler divergences to the targets and smaller importance-weight variances than a single-component approximation which uses more Monte Carlo samples to estimate the objective."
> >
> > * "Note that minimizing the denominator in Eq. (6) corresponds to diversifying the mixture distribution." --> "Note that maximizing Eq. (6) w.r.t. the variational parameters will promote the components to diversify and explore regions of the latent space which correspond to high-probability regions measured by $p$. That is, samples drawn from component $s$ that result in high values in the numerator will indicate that the $s$-th component covers a part of the latent space corresponding to a high-probability region, attracting the other components to this region, while the denominator penalizes similarities among the components, promoting the other components to spead out and find other important parts of the latent space."
> >
> > * Regarding the disjoint SBN parameter sets, this is now mentioned also in Sec. 3.1.1, and Appendix F now reads "Additionally, using shared parameters for the mixture components can also be utilized, if the practitioner is running on a limited memory budget. However, shared weights would violate the assumption of disjoint SBN parameter sets in Appendix F. Nonetheless, devising clever modifications [...]"
> >
> > * "(ii) as the second term is negated, maximizing it corresponds to diversifying the mixture distribution w.r.t. $\phi_i$" --> "$(ii)$ as the second term is negated, maximizing it promotes component $i\neq s$ to be dissimilar from component $s$, diversifying the mixture."
> >
> > * "the increased model flexibility and promotion of exploration translates into better marginal log-likelihood estimates" --> "the increased model flexibility and promotion of exploration translates into lower-variance estimators of the marginal log-likelihood"
> >
> > Once again, we are sincerely grateful for your feedback and careful comments.

---

> > > ### Comment · Reviewer_LrA3 · 2024-10-24
> > > **Reviewer response**
> > >
> > > Thank you for the above updates.
> > >
> > > While I believe it is important for authors to archive trained models until publication to address reviewer requests for figures or additional analysis, I understand that unforeseen circumstances can arise. Given the efforts made by the authors and the improvements already implemented in the updated manuscript, I am willing to overlook this issue. Based on our previous discussions and the revisions, I believe the paper makes a valuable contribution and will be of interest to the readers of TMLR. Therefore, I recommend its acceptance.

---

> > > > ### Author Response · Authors · 2024-10-25
> > > > **Thank you**
> > > >
> > > > We are very grateful for your consideration, and pleased that you recommend accept at this point. Again, thanks for your thorough comments and analyses throughout the review process! You efforts have been very valuable.

---

### Review · Reviewer_5pL4 · 2024-10-13

**Summary Of Contributions:**

This paper proposed a variant of variational Bayesian phylogenetic inference that uses mixture of SBNs instead of a single SBNs for tree topology posterior approximation. The authors show that certain distribution of tree topologies can not be properly learned via a single SBN. Stochastic gradient estimators (following the VIMCO method) have been derived. Experiments on a toy hierarchical model and real data Phylogenetic posterior estimation tasks demonstrate the effectiveness of the proposed method.

**Audience:**

Yes

**Broader Impact Concerns:**

No broader impact concerns are needed so far.

**Claims And Evidence:**

Yes

**Requested Changes:**

No requested changes.

**Strengths And Weaknesses:**

The authors have addressed the concerns raised in the previous review round effectively in this revised version of the paper. The modifications have significantly improved the clarity and addressed the key issues I highlighted earlier. Based on these improvements, I believe the paper is now in good shape for acceptance.

---

### Review · Reviewer_9pfv · 2024-10-18

**Summary Of Contributions:**

The authors propose a mixture model of tree topologies to enhance parameter learning in Variational Bayesian Phylogenetic Inference (VBPI). Additionally, they derive a VIMCO estimator for mixtures.

**Audience:**

Yes

**Claims And Evidence:**

Yes

**Requested Changes:**

I would suggest adjusting the position of the legend in Fig. 3 or removing the bounding box for better clarity.

**Strengths And Weaknesses:**

I have reviewed the previous submission of this paper, and I’m pleased to see that most of my earlier comments have been addressed, including the inclusion of running times, clarification on the use of pre-computed topologies, and softening of the language. Given these improvements, I would support acceptance.

---

### Author Response · Authors · 2024-10-25

We would like to thank the reviewers and the action editor for their hard work throughout these two rounds of submissions. We are very pleased with the constructive and actionable feedback that we have received from you, and our work has been greatly improved. Again, thank you all!

---

### Decision · Action_Editor_J9XE · 2024-11-11

**Recommendation:** Accept as is

**Comment:**

This paper is a resubmission to TMLR. The three reviewers found that the submission addressed all the raised concerns, and therefore the paper is ready for acceptance. See details of the previous submission [here](https://openreview.net/forum?id=yuhG3VHK5f).

The paper is well written, and its technical correctness has been significantly improved with respect to the previous version.

**Audience:**

All three reviewers agree on the useful contributions of the paper, which tackles the challenging problem of approximate posterior inference, making the manuscript of interest for the Bayesian community.

**Claims And Evidence:**

This work makes useful technical contributions in the context of VBPI and VIMCO using mixtures. The evaluations are convincing, and they show some preference for the proposed approach on certain metrics of interest.